# Rotavirus A in Domestic Pigs and Wild Boars: High Genetic Diversity and Interspecies Transmission

**DOI:** 10.3390/v14092028

**Published:** 2022-09-13

**Authors:** Dragan Brnić, Daniel Čolić, Valentina Kunić, Nadica Maltar-Strmečki, Nina Krešić, Dean Konjević, Miljenko Bujanić, Ivica Bačani, Dražen Hižman, Lorena Jemeršić

**Affiliations:** 1Virology Department, Croatian Veterinary Institute, Savska cesta 143, 10000 Zagreb, Croatia; 2Institute for Research in Biomedicine (IRB Barcelona), The Barcelona Institute of Science and Technology, Baldiri Reixac 10, 08028 Barcelona, Spain; 3Laboratory for Electron Spin Spectroscopy, Division of Physical Chemistry, Ruđer Bošković Institute, Bijenička cesta 54, 10000 Zagreb, Croatia; 4Faculty of Veterinary Medicine, University of Zagreb, Heinzelova 55, 10000 Zagreb, Croatia; 5Animal Feed Factory Ltd., Dr. Ivana Novaka 11, 40000 Čakovec, Croatia; 6Belje Agro-Vet plus Ltd., Kokingrad 4, Mece, 31326 Darda, Croatia; 7Rusagro, LLC “Tambovsky bacon”, Bazarnaya 104, 392036 Tambov, Russia

**Keywords:** rotavirus A, domestic pig, wild boar, genotype, genetic diversity, molecular epidemiology, phylogenetic analysis, interspecies transmission, Croatia

## Abstract

Rotavirus A (RVA) is an important pathogen for porcine health. In comparison to humans, RVA in domestic animals and especially in wildlife is under researched. Therefore, the aim of the present study was to investigate the prevalence, genetic diversity, molecular epidemiology and interspecies transmission of RVA in domestic pigs and wild boars. During the three consecutive RVA seasons (2018–2021) we collected 445 and 441 samples from domestic pigs and wild boars, respectively. Samples were tested by real-time RT-PCR, and RVA-positive samples were genotyped in VP7 and VP4 segments. Our results report an RVA prevalence of 49.9% in domestic pigs and 9.3% in wild boars. Outstanding RVA genetic diversity was observed in VP7 and VP4 segments, especially in domestic pigs exhibiting a striking 23 different RVA combinations (G5P[13] and G9P[23] prevailed). Interspecies transmission events were numerous between domestic pigs and wild boars, sharing G3, G5, G6, G9, G11 and P[13] genotypes. Furthermore, our data indicate that such transmission events involved even bovines (G6, P[11]) and, intriguingly, humans (G1P[8]). This study contributes to the basic knowledge that may be considered important for vaccine development and introduction, as a valuable and currently missing tool for efficient pig health management in the EU.

## 1. Introduction

Rotaviruses (RV), the species *Rotavirus A* (RVA) in particular, represent a major healthcare burden worldwide [1]. They are a significant enteric pathogen in intensive animal farming as well, especially among younger animals [2]. The focus of the research community has been primarily oriented towards human rotaviruses, with 30 and 100 times fewer genotyped RVA strains of bovine and swine origin, respectively [3]. The knowledge gap is even wider on rotaviruses circulating in wildlife [4].

The family *Reoviridae* and the genus *Rotavirus* encompass a diverse group of rotavirus species; *Rotavirus A–D* and *Rotavirus F–J* [5]. The RVA is by far the species with the most significant impact on human and animal health [6]. These dsRNA viruses possess a genome consisting of 11 segments, which are enclosed in a triple-layered capsid [7]. The rotavirus nomenclature is binomial and based on the two genomic segments encoding two outer capsid structural proteins, VP7 and VP4, which define the G and P genotypes, respectively. However, the nomenclature based on the genotype assignment for all 11 segments has been developed [8]. Currently, there are at least 41 G and 57 P genotypes recognized by the Rotavirus Classification Working Group (RCWG) [9].

Direct interspecies transmission and transmission coupled with reassortment are considered to be two major routes for rotaviruses to cross the host barrier [10]. The segmented form of the rotavirus genome is a prerequisite for the occurrence of new chimeric reassortant strains, very often of human and animal origin with many examples involving domestic animals [11]. However, the importance of wildlife in such interspecies transmission events might be underestimated since the research on that topic has largely been neglected by scientists worldwide.

Rotaviruses provide an everlasting challenge for pig health management due to their ubiquitous nature and high resistance in the environment [12]. The disease they cause is usually self-limiting gastroenteritis, which can be fatal in young piglets due to dehydration, with outbreaks being especially severe in intensive farming systems [2]. The prevalence of RVA in clinically affected and asymptomatic pigs ranges between 3.3 and 67.3%, without evidence of seasonality, but with spatio-temporal variations and the re-emergence of certain genotypes [13]. Despite being under-researched, RVAs circulating in domestic pigs exhibit high diversity with at least 50 known genotype combinations [6]. Domestic pigs are considered to be the origin of some RVA genotypes circulating in humans, such as G9 and G12 genotypes [11]. Moreover, it is considered that human Wa-like and porcine RVAs share a common ancestor [14]. Apart from reducing RVA build-up through strict hygiene measures, the importance of boosting lactogenic immunity with vaccination remains the most important tool for confronting RVA infection adverse outcomes [2]. Nevertheless, currently, there is no authorized vaccine against porcine RVA in the European Union, yet some pig producers rely on vaccine importation from the USA [15]. In light of the high genetic diversity of porcine RVAs, there are concerns regarding vaccine efficacy on heterologous strains [13,15].

RVA in wild boars, to the best of our knowledge, has been scarcely researched so far with only two papers dealing with molecular epidemiology and genetic diversity of circulating strains. Both advocate interspecies transmission of RVAs between domestic pigs and wild boars [16,17]. In addition, the close phylogenetic relatedness to certain RVA strains detected in humans was described [16,17].

The aim of the present study was to reveal the concurrent prevalence, molecular epidemiology, genetic diversity and possible interspecies transmission events of RVA strains circulating in domestic pig and wild boar populations in Croatia during three consecutive RVA seasons.

## 2. Materials and Methods

### 2.1. Sampling

From 2018 to 2021 (covering rotavirus seasons 2018/2019, 2019/2020 and 2020/2021), we sampled 445 domestic pigs and 441 free-living wild boars (*Sus scrofa*). All animals were sampled only once. Most of the samples (98.2% of domestic pigs and 78.9% of wild boars) were collected during the autumn/winter months (October to March). Sampled animals originated from 10 counties (seven for domestic pigs and eight for wild boars) of the Continental Croatia (Pannonian Croatia, Northern Croatia and City of Zagreb according to the NUTS-2 classification) and in one county (Split-Dalmatia County: domestic pigs) of the Adriatic Croatia (Figure 1). Domestic pigs included in the present study were locally bred on 24 small backyard holdings (*N* = 276) and eight large holdings (*N* = 169). Holdings breeding imported weanling and fattening pigs were not included in the present study. Some holdings were visited for sampling more than once, but different animals were sampled each time. Wild boars were sampled after regular hunting in 15 hunting areas located in eight counties (Figure 1).

All samples were taken from individual animals by rectal swabs (young age categories of domestic pigs) or by a plastic scoop attached to a container’s lid when fecal samples or intestinal content were sampled (wild boars). The age, gender and status of diarrhea were registered at the time of sampling. Domestic pigs were divided into four age groups: suckling piglets (<28 days; *N* = 231), weanling pigs (29–84 days; *N* = 177), fattening pigs (>85 days, *N* = 28) and sows (*N* = 9). On the other hand, three age groups were defined for wild boars: <1 year (*N* = 151), 1–2 years (*N* = 135) and >2 years (*N* = 155). The gender was reported for 385 domestic pigs (178 females and 207 males) and 440 wild boars (223 females and 217 males). Diarrhea was registered in 165 domestic pigs (37.1%) and in only eight wild boars (1.8%). Other animals were observed as healthy regarding their gastrointestinal tract: 280 domestic pigs (62.9%) and 433 wild boars (98.2%). All samples were transferred to the laboratory maintaining the cold chain and stored at −20 °C until further use.

### 2.2. RNA Extraction and Real-Time RT-PCR

RNA was extracted from the supernatant of 20% *w/v* fecal suspension, which was prepared using Medium 199 (Sigma Aldrich, St. Louis, MO, USA), vortexed and centrifuged at 14,000× *g*. The RNA extraction procedure was performed on the KingFisher™ Flex purification system (ThermoFisher Scientific, Waltham, MA, USA) using the MagMAX™ CORE Nucleic Acid Purification Kit (ThermoFisher Scientific, Waltham, MA, USA) by following the manufacturer’s instructions (complex workflow). The exogenous Internal Positive Control (IPC) RNA, Xeno™ RNA Control (ThermoFisher Scientific, Waltham, MA, USA), was added to each sample (2 µL) to supervise the appearance of potential PCR inhibitors. The extracted RNA was stored at −80 °C if not processed immediately.

Detection of the RVAs dsRNA was performed by real-time RT-PCR, which amplifies the fragment of VP2 segment of different RVA genotypes infecting humans and domestic animals [18]. Nevertheless, this protocol was previously successfully applied for RVA detection in wildlife-related research [19,20]. The reaction mixture setup and thermal cycling conditions were described previously [20]. The runs were performed on a Rotor-Gene Q or QIAquant 96 5plex (Qiagen, Hilden, Germany). If inhibition was observed, the samples were diluted at 1:5 and retested.

### 2.3. VP7 and VP4 Genotyping

All VP2 positive samples were subjected to genotyping in order to define the G genotype (VP7) and P genotype (VP4). Several approaches were applied to overcome possible bottlenecks due to the RVA strain genetic diversity.

The VP7 genotyping was performed with a combination of VP7 Beg9 and VP7 End9 primers [21] in the first round of RT-PCR followed by the nested PCR using VP7-up2 and VP7-down3 primers [22]. The next approach was the RT-PCR using VP7-F and VP7-R primers, followed by the seminested PCR using VP7-F and VP7-RINT primers [23] if the result of the first RT-PCR reaction was negative. In some cases, we applied primers N-VP7F1 and N-VP7R1 in the first round of RT-PCR, and primers N-VP7F2 and N-VP7R2 in the nested PCR. These primer sets were designed for samples containing low RVA load [24].

The VP4 genotyping was a combination of three different approaches as well. One approach was the application of VP4-HeadF and VP4-1094R2 primers in the RT-PCR followed by the seminested PCR using VP4-HeadF and VP4-887R primers [22]. The other one was a combination of VP4_1-17F and VP4R_DEG primers in the RT-PCR reaction [25]. The last approach consisted of N-VP4F1 and N-VP4R1 in the RT-PCR, and N-VP4F2 and N-VP4R2 in the nested PCR [24].

All RT-PCR reactions were conducted with the utilization of SuperScript™ III One-Step RT-PCR System with Platinum™ Taq DNA Polymerase (ThermoFisher Scientific, Waltham, MA, USA). For the nested or seminested PCR, GoTaq^®^ G2 Hot Start Colorless Master Mix (Promega, Madison, WI, USA) was utilized. Primer concentrations and annealing temperatures used in each RT-PCR and nested or seminested PCR reaction were as recommended by the corresponding article. Other conditions, related to the reaction mixture setup and thermal cycling, were as recommended by the reagent’s manufacturer. Each reaction started with the initial dsRNA denaturation step at 95 °C for 5 min in which extracted RNA was combined with the respective forward primer and PCR grade water. Hereafter, the remaining reagents were added to the reaction mixture, which was run on the ABI 9700 GeneAmp thermal cycler (Applied Biosystems, Foster City, CA, USA) or Biometra TRIO (Analytic Jena, Jena, Germany). Reactions were visualized on the QIAxcel Advanced System for capillary electrophoresis using the QIAxcel DNA Screening kit (Qiagen, Hilden, Germany).

### 2.4. Genotype Assignment and Phylogenetic Analysis

The purification of RT-PCR and PCR products was performed by ExoSAP-IT™ PCR Product Cleanup Reagent (ThermoFisher Scientific, Waltham, MA, USA) or Monarch DNA Gel Extraction Kit (New England Biolabs, Ipswich, MA, USA) as described previously [20]. Subsequently, the samples were subjected to direct Sanger sequencing in both directions using the services of Macrogen Europe (Amsterdam, the Netherlands). The RVA genotypes of VP7 and VP4 segments were assigned by following previously defined cutoffs [8] and using BLAST search (https://blast.ncbi.nlm.nih.gov/Blast.cgi, accessed on 24 February 2022) in combination with the ViPR tool [26] available at https://www.viprbrc.org/ (accessed on 24 February 2022).

The phylogenetic analysis was performed with the selected sequences of adequate quality and length, which represented a certain genotype, lineage and origin. In the analysis, we included the selected number of referent sequences obtained from the GenBank. The MUSCLE algorithm was utilized for the purpose of building a multiple sequence alignment, preceding a phylogenetic analysis conducted with the maximum-likelihood (ML) method. Two substitution models with the lowest BIC score were applied: T92+G+I (all VP7 and VP4 sequences of P[13], P[23] and P[32] genotypes) and T92+G (VP4 sequences of P[6]–P[8] and P[11] genotypes). The branching support of the ML tree was assessed by the bootstrap analysis with 1000 repetitions. These analyses were performed in MEGA11 software [27]. The phylogenetic trees were visualized and annotated using iTOL (version 6.5.8) [28]. The nucleotide pairwise identity matrix and graphical overview of the temporal distribution of RVA genotypes circulating in domestic pigs were calculated in R using the bio3d package, ggplot2 and Scatter Pie Plot [29,30,31,32]. The lineage designation of a certain genotype was set by the previously recommended classification for G1, G2, G3, G4, G6, G9, P[6] and P[8] genotypes [19,33,34,35,36,37,38,39] due to their high frequency in humans (G1-G4, G9 and P[8]) or due to the close phylogenetic relatedness observed between human and animal RVA strains (G6 and P[6]). As a result of the general inconsistency in the nomenclature and/or the absence of consensus in the lineage demarcation, we opted not to define lineages for other G and P genotypes reported within the present study.

RVA sequences characterized in the present study are deposited to the GenBank under accession numbers OL440064-OL440111, ON017591-ON017611, ON647404-ON647430, ON721080-ON721102 and OP136969.

### 2.5. Statistical Analysis

Descriptive statistics (prevalence) and comparison of the type of holding (farm/backyard), age and gender in affected (diarrheic) and non-affected animals (non-diarrheic) were performed in SYSTAT 13 For Windows© Version No.13.2, Systat Software, Inc. 2017. For the categorical data analysis, χ^2^ test and log-linear model (LLM) were used. For all analyses, *p* < 0.05 was considered statistically significant.

## 3. Results

### 3.1. RVA Prevalence in Domestic Pigs and Wild Boars

Our results demonstrate that RVA is a highly prevalent pathogen in domestic pigs with 49.9% (222/445; 95% CI, 45.1–54.6%) positive samples. It was significantly more prevalent (*p* < 0.00001) in pigs bred on large holdings (115/169; 68.1%) compared to small backyard holdings (107/276, 38.8%). Moreover, the prevalence was significantly (*p* < 0.00001) higher in diarrheic animals (118/165, 71.5%) compared to those that were healthy, i.e., without gastrointestinal symptoms (104/280, 37.1%). On the other hand, statistical significance was not found in the RVA prevalence between suckling (113/231, 48.8%) and weanling pigs (93/177, 52.5%) (*p* < 0.468; older age categories were excluded from the analysis due to the small sample number) and between females (83/178, 46.6%) and males (99/207, 47.8%) (*p* < 0.895). All large holdings (*N* = 8) and 20 out of 24 small backyard holdings were positive for RVA in at least one sampled animal.

Wild boars were mostly RVA negative since only 41 of 441 tested animals were positive, which gives an RVA prevalence of 9.3% (95% CI, 6.8–12.4%). All positive wild boars were within the healthy group. The age group was not a significant factor (*p* < 0.341) for the RVA prevalence in wild boars since RVA was detected in 11.9% (18/151) of animals under one year of age, in 8.9% (12/135) of those between one and two years of age, and in 7.1% (11/155) of animals older than two years of age. Similarly, gender was not a significant factor (*p* < 0.291) either, since RVA was detected in 10.7% (24/223) of female compared to 7.8% (17/217) of male wild boars.

RVA prevalence on the county level for domestic pigs and wild boars is shown in Figure 1.

The observed Cq values for the RVAs detected in domestic pigs were in 55% (122/222) and 45% (100/222) under and over 32, respectively. In wild boars, a Cq lower than 32 was observed in 36.6% (15/41) of samples compared to 63.4% (26/41) of samples in which it was higher than 32. The result of IPC amplification reveals the general absence of PCR inhibitors in the majority of samples of domestic pigs (99.3%, 442/445) and wild boars (95.7%, 422/441). Those samples that were retested in 1:5 dilutions (22 in total: 3 and 19 in domestic pig and wild boar sample sets, respectively) were mostly RVA negative (20/22). Only two RVA positives were detected in diluted samples that originated from domestic pigs.

### 3.2. VP7 and VP4 Genotype Diversity in RVA Strains Circulating in Domestic Pigs and Wild Boars

The genotyping procedure, described in Materials and Methods, was at least partially successful (G or P genotype was defined) in 176 out of 222 (79.3%) RVA-positive samples of domestic pigs and in 24 out of 41 (58.5%) RVA-positive samples of wild boars.

In domestic pigs, we determined the circulation of eight and seven different G and P genotypes, respectively. The G genotype was detected in 163 samples. Namely, G9 (*N* = 42, 25.8%), G5 (*N* = 40, 24.5%), G3 (*N* = 30, 18.4%), G1 (*N* = 19, 11.7%), G4 (*N* = 16, 9.8%), G2 (*N* = 14, 8.6%), G6 (*N* = 1, 0.6%) and G11 (*N* = 1, 0.6%). The G3, G5 and G9 genotypes combined were the most dominant genotypes (68.7%). The P genotype was determined in 140 samples as follows: P[13] (*N* = 60, 42.9%), P[23] (*N* = 55, 39.3%), P[8] (*N* = 8, 5.7%), P[6] (*N* = 6, 4.3%), P[32] (*N* = 5, 3.6%), P[7] (*N* = 3, 2.1%) and P[11] (*N* = 3, 2.1%). The most dominant genotypes were P[13] and P[23] with a combined share of 82.1%. The geographical distribution of RVA G and P genotypes circulating in domestic pigs was shown in Figure 1 and the temporal distribution in Figure 2. Not all genotypes were present in all three RVA seasons covered by this study; for instance, the G1 and P[8] genotypes were detected only in the season 2020/2021, G6 in the season 2018/2019 and G11 in the season 2019/2020. The G/P genotype combination was defined for 127 RVA strains in domestic pigs. In total, there were 23 different genotype combinations with G5P[13] and G9P[23] being the most prevalent (49.6%). Genetic diversity was the highest on large holdings where we detected up to six different G and six different P genotypes in one holding. On the contrary, the genetic diversity was lower on backyard holdings with up to three and two different G and P genotypes per holding, respectively. If we consider only one sampling time point per holding, the observed genetic diversity of circulating RVA strains was at four different G and six different P genotypes on large holdings and up to three different G and two different P genotypes on small backyard holdings.

In wild boars, the genetic heterogeneity of circulating RVA strains was lower with five different G genotypes and only one P genotype detected during three consecutive RVA seasons. The G genotype was detected in 23 and the P genotype in 13 wild boar samples. The most frequent G genotype was G3 (*N* = 12), followed by G5 (*N* = 4), G9 (*N* = 3), G6 (*N* = 2) and G11 (*N* = 2). All 13 samples with the successful detection of the P genotype, belonged to the P[13] genotype. The G/P genotype combination was defined for 12 RVA strains, namely, G3P[13] (*N* = 8), G5P[13] (*N* = 2), G9P[13] (*N* = 1) and G11P[13] (*N* = 1). The geographical distribution of RVA G and P genotypes detected in wild boars was shown in Figure 1, and the temporal distribution was described in Section 3.3.

Even though successful and reliable genotype definitions can be achieved for shorter sequences [23,24], a small portion of G genotypes and all P genotypes did not meet the previously defined requirements [8]; hence, these genotypes are to be considered as candidate genotypes of already established ones.

### 3.3. The Results of Phylogenetic Analysis of RVA Strains in Domestic Pigs and Wild Boars

#### 3.3.1. VP7 Genotyping

##### G1

The RVA strains of this common human genotype were detected in season 2020/2021 (Figure 2) on four domestic pig holdings in three counties (Figure 1). All these porcine G1 strains were closely phylogenetically related since they share 99.9–100% nucleotide (nt) sequence identity and 99.6–100% amino acid (aa) sequence identity. They branched within lineage I (Figure 3A), which is a common lineage for the majority of human RVA strains with which they share 96.8–98.5% on the nt and 97–98.1% on the aa level. The G1 genotype was detected in combination with P[7], P[8], P[11] and P[23].

##### G2

The G2 RVA strains detected in domestic pigs within the present study were placed in the potentially novel lineage (Figure 3B), since they share less than 90% nt identity and 91.8–93.5% aa identity with the closest porcine RVA strains. Nevertheless, they are more closely related to porcine than to human strains (Figure 3B). If we observe only G2 strains from the present study, they share a high resemblance with 98.7–99.9% and 98.9–100% on the nt and aa levels, respectively. These G2 strains were detected on seven holdings in six counties (Figure 1), covering all three sampling seasons (Figure 2). They come in combination with P[13], P[23] and P[32] genotypes.

##### G3

Genotype G3 was the third most abundant genotype circulating in domestic pigs, and the most prevalent genotype in wild boars. These RVA strains form a diverse group of sequences, sharing 85.9–100% sequence identity on the nt level and 93.1–100% on the aa level. Two clusters of RVA strains detected in the present study can be recognized; the first one with mixed strains originating from domestic pigs and wild boars and the second one observed only in domestic pigs (Figure 3A). Within the first cluster, RVA strains from domestic pigs and wild boars share sequence identity in the range between 90.3 and 97.5% on the nt and 94.7–98.5% on aa level. The second cluster might represent a distinct lineage since sequence identities were lower than 89% (Figure 3A). All RVA strains from our study were quite clearly separated from the two lineages circulating in humans, lineage I representing classical human RVA strains, and lineage IX representing equine-like RVA strains (Figure 3A). The phylogenetically closest RVA strains to Croatian autochthonous strains were those detected in domestic pigs, sharing the highest 91.7% nt and 96.9% aa identity with the Slovakian RVA strain TOPC23 (MN203555). The strains of G3 genotype in domestic pigs were detected in eight holdings and three counties (Figure 1) during all three sampling seasons (Figure 2). They circulated in combination with six out of seven P genotypes detected in our study (P[6]–P[8], P[11], P[13] and P[23]). On the other hand, wild boar RVA strains were detected in six hunting grounds and four counties (Figure 1) during all sampling seasons, as well. More precisely, three, eight and one G3 strain in the 2018/2019, 2019/2020 and 2020/2021 seasons, respectively. Since the circulation of only one P genotype in wild boars was determined, the G3 strains combined solely with the P[13] genotype.

##### G4

RVA strains of this genotype were detected only in domestic pigs in all three sampling seasons (Figure 2). They were circulating in seven holdings and three counties (Figure 1). These strains were quite diverse, clustering in three separate groups (Figure 3B), which may represent three different lineages due to the low nucleotide sequence identity, ranging between 82.6 and 85.8%. Nevertheless, if the lineage designation described in Wandera et al. [39] was applied, these strains would cluster within lineage VI, which is in contrast with the demarcation threshold of previously established lineages (I–V). The G4 strains detected in the current study, share a high phylogenetic relationship with the different RVA strains detected in domestic pigs, wild boars and humans (Figure 3B). The latter are considered to be a zoonotic spillover [40]. The G4 genotype was found in combination with P[6], P[13] and P[23] RVA strains.

##### G5

This typically porcine RVA genotype was confirmed to circulate in both species: domestic pigs and wild boars. It was present during all three sampling seasons in domestic pigs (Figure 2) and during the last two sampling seasons in wild boars (three strains in the season 2019/2020 and one in the season 2020/2021) The RVA strains of this genotype clustered in three groups (Figure 3A) potentially representing three distinctive lineages since the nucleotide identity is in the range between 83 and 88.4%. The first group includes strains from domestic pigs and wild boars, while the other two were detected only in domestic pigs. The RVA strains of domestic pigs and wild boars described in the present study are highly phylogenetically related (Figure 3), sharing 95.5–97.4% and 96.7–98.9% on the nt and aa levels, respectively. Overall, this genotype was detected in the largest number of holdings (*N* = 14) from six counties (Figure 1), when domestic pigs are considered. Within the wild boar population, the G5 genotype was detected in two hunting grounds from two counties (Figure 1). When we look at combinations with the P genotype, they were less diverse compared to the other prevalent G genotypes. Accordingly, in domestic pigs, we observed only combinations with P[13] and P[23] genotypes, whereas G5P[13] was the sole combination observed in wild boars.

##### G6

The G6 genotype, a common RVA genotype in cattle, was detected in one domestic pig and two wild boars during the season 2018/2019 (Figure 2), each from a different county (Figure 1). The strain detected in the domestic pig was found in combination with the P[13] genotype. The length of sequences derived from wild boars was not informative enough for the inclusion of these strains into the phylogenetic analysis. Nevertheless, the genotype assignment was reliable [24]. The sequence detected in a domestic pig was closely related to a bovine RVA strain from Hungary and a bovine-like RVA strain detected in a child from Slovenia (97.7%/99.2% on the nt/aa level) and clustered within the lineage V (Figure 3B).

##### G9

This genotype was the most prevalent genotype in domestic pigs and the third most prevalent genotype in wild boars. It was detected in 10 domestic pig holdings located in three counties, whilst wild boar G9 strains were detected in three hunting grounds from three counties (Figure 1). It circulated during each sampling season of domestic pigs (Figure 2), and in the first (2018/219, *N* = 2) and third (2020/2021, *N* = 1) sampling seasons of wild boars. Despite being the most prevalent, this genotype was found in combination with only P[13], P[23] and P[32] genotypes. In wild boars, it combined with P[13], the sole P genotype detected in that species. All G9 RVA sequences from the present study were closely related (92.2–99.5%/94.7–100% on the nt/aa level) and showed high identities (95.3%/95% on the nt/aa level) among wild boar and domestic pig strains (Figure 3B). Together with the Italian and Belgian domestic pig strains, they form a potentially novel lineage as already proposed [41]. As a matter of interest, the strain derived from the red fox, described in our concurrent study [20], phylogenetically clustered together with porcine strains from this study sharing high sequence identities with several strains (99.7/99.5% on the nt/aa level) (Figure 3B). Human and porcine G9 strains from the current study were clearly phylogenetically distinct (Figure 3B).

##### G11

The low prevalent, porcine-related genotype was confirmed in only one domestic pig and two wild boars during seasons 2019/2020 and 2020/2021, respectively, and geographically originated from two counties. Indeed, the majority of related referent RVA sequences were of porcine origin with an evident close phylogenetic relationship (Figure 3B). Our G11 RVA strains detected in a domestic pig and wild boars were clustered in clearly distinguished lineages (Figure 3B) sharing only 86% nucleotide sequence identity. Likewise, for the G9 genotype, it is interesting to note that this genotype was detected in a red fox during our concurrent study [20]. However, the red fox G11 strain clusters in a separate lineage (Figure 3B) considering the low 86.4% and 87.7% nucleotide identities it shares with wild boar and domestic pig strains, respectively.

#### 3.3.2. VP4 Genotyping

##### P[6]

The P[6] genotype was not among the most prevalent VP4 RVA genotypes in domestic pigs during the present study. However, the importance of these strains lies in the close phylogenetic connection to human P[6] strains from Hungary and Slovenia within lineages IV and V, respectively (Figure 4A). With these human strains, they share nt/aa identities of 94.1–97.8%/90.3–97.7%. These human RVA strains are reported to represent the event of porcine-to-human zoonotic transmission [33,40]. The six P[6] strains from the present study originated in two counties (Figure 1) and four holdings and were detected in all three sampling seasons (Figure 2). These strains were detected in combination with G3 and G4 genotypes.

##### P[7]

This primarily porcine genotype was detected in three domestic pigs in the last sampling season (2020/2021) in two holdings from two counties (Figure 1). Our strains shared the highest similarity with the wild boar strain from the Czech Republic (95.9%/92.6% on the nt/aa level) (Figure 4A). However, none of the wild boar samples in our study presented this genotype. These three P[7] strains were detected in combination with human-like G1 and G3 genotypes.

##### P[8]

Likewise the P[7] genotype, this genotype emerged in the 2020/2021 season (Figure 2). It is considered to be the most common genotype in humans; hence, the expected high sequence identities (up to 99.5/98.6 on the nt/aa level) were observed with human strains of lineage III (Figure 4A). Nevertheless, the close phylogenetic relatedness was identified with porcine strains detected in the United Kingdom and Taiwan (Figure 4A). When we look at the mutual relatedness of our P[8] strains, they share 96.2–100% and 93–100% on the nt and aa levels, respectively. These RVA strains emerged in three holdings and three counties (Figure 1) and were combined with human-like G1 and G3 genotypes.

##### P[11]

The P[11] genotype is considered to be one of the most frequent bovine genotypes. However, in the present study, we detected it in three domestic pigs from two counties (Figure 1). These strains elicited a high sequence identity with different RVA strains circulating in cattle (95.3–98% on the nt and 93.9–99.5% on the aa level) (Figure 4A). Our previous investigation reports the presence of this genotype in red foxes, which is closely phylogenetically related to domestic pig strains from the present study (97.1%/98.1% on the nt/aa level) (Figure 4A). The P[11] genotype was determined in combination with human-like G1 and G3 genotypes.

##### P[13]

One of the two most numerous P genotypes characterized in the present study was detected in 16 domestic pig holdings located in five counties (Figure 1) in all three sampling seasons (Figure 2). Moreover, this genotype was detected in wild boars in five hunting grounds from three counties during all sampling seasons (one, nine and three strains in the 2018/2019, 2019/2020 and 2020/2021 seasons, respectively). It was detected in combination with seven different G genotypes (all except human-like G1) in domestic pigs and four different G genotypes (all except G6) in wild boars. The phylogenetic analysis revealed that P[13]-circulating RVA strains have the highest intragenotype diversity of all genotypes detected in the present study. In total, six clusters (Figure 4B) were defined: one cluster with wild boar strains (DS302-OB as representative strain), three clusters with strains from domestic pigs (S95-MZ, S421-MZ and S70-VS) and two clusters of combined RVA strains originating in both species (S22-MZ/DS229-Zag and S348-OB/DS327-Zag). The highest sequence identity between these clusters was 88.9% and 89.3% on the nt and aa levels, respectively. It is evident that the closest relatives to our P[13] are RVA referent strains originating from domestic pigs and wild boars (Figure 4B). The interesting observation is the close resemblance of two red fox RVA strains described in our previous study [20] and strains/clusters from domestic pigs S70-VS and S95-MZ (nt sequence identity approx. 95%) (Figure 4B). The phylogenetic grouping of domestic pig and wild boar strains within two clusters was confirmed with high sequence identities of up to 98.6%/98.1% on the nt/aa level in cluster S22-MZ/DS229-Zag and 97.4%/96.8% on the nt/aa level in cluster S348-OB/DS327-Zag.

##### P[23]

The second most numerous P genotype, detected only in domestic pigs, showed a significantly lower genetic heterogeneity since all sequences are grouped within a single cluster (Figure 4B) with identities ranging between 90.1–99.8% on the nt and 92.9–99.5% on the aa level. This genotype was detected in 11 holdings in five counties (Figure 1) during all three seasons (Figure 2). Furthermore, it was found circulating in combination with six G genotypes (G1-G5, G9). Interestingly, one fox strain from our previous study [20] was phylogenetically closely related to domestic pig RVA strains (Figure 4B) sharing up to 99.7/100% identity on the nt/aa level.

##### P[32]

This rare genotype was confirmed circulating in Croatia, but with restricted regional importance since it was detected only in five strains originating in three holdings from two neighboring northernmost counties (Figure 1). It circulated in all three seasons (Figure 2) and came in combination with G2 and G9 RVA strains. The closest P[32] strains from the GenBank were those from the UK (Figure 4B) and Switzerland (not shown) sharing less than 89% identity on the nt level and 90% on the aa level. The evident separate clustering indicates a possible circulation of a distinct lineage.

## 4. Discussion

The present study brings a comprehensive concurrent investigation of the prevalence, molecular epidemiology and genetic diversity of circulating RVA strains in domestic pigs and wild boars during three consecutive RVA seasons. The concurrent spatio-temporal study of RVA strains circulating in certain reservoir species and the environment has previously been recognized as important in order to draw relevant conclusions on their prevalence and health impact [3,42]. Furthermore, this is one of the rare studies on RVAs in domestic pigs in this part of the world (South East Europe), and only the third in general, to the best of our knowledge, where RVAs in wild boars have been considered.

The prevalence in domestic pigs presented in the current study is relatively high (49.9%), similar to previously described studies in Spain [15], Italy [43] and the USA [44]. Most certainly, the predominantly represented young age categories of suckling and weanling pigs, favor the more frequent circulation of RVAs. Moreover, the used method of detection has a substantial influence on the final result. Similar to our study, Spanish, Italian and US researchers [15,43,44] used the real-time RT-PCR based method, while in contrast, Taiwanese researchers used the Enzyme Immunoassay as screening and end-point RT-PCR as the confirmatory method [45], which finally resulted in a significantly lower prevalence that is incomparable. Our results indicate that there is no significant difference in the prevalence between the suckling and weanling age categories and between females and males, but the difference in RVA prevalence was significant in diarrheic compared to healthy animals, and in those bred on large holdings compared to the small backyard holdings. RVAs are known causative agents of diarrhea in mammals [2], and our research brings more to that knowledge.

Data on the RVA significance in wild boars have been rather scarce so far with only two available reports from Japan [16] and the Czech Republic [17]. Our study is the most comprehensive to date, encompassing a sample set of 441 animals. It is also noteworthy that the sampling was performed in parallel with domestic pigs, which provides a temporal component important for relevant phylogenetic comparisons. RVA prevalence in the present study (9.3%) was higher compared to those two previous studies, primarily due to having a different approach to RVA detection. We applied the real-time RT-PCR compared to the conventional end-point RT-PCRs applied by others [16,17], which are usually less sensitive. The method we implemented has been previously successfully applied in RVA-related research on domestic animals and wildlife [18,19,20]. Nevertheless, the unknown range of VP2 genotypes that this method detects, and the fact that the assay design was limited to only human strains of C1 and C2 genotypes, might have underestimated the prevalence in both species. Moreover, the prevalence in wild boars might be even higher, since, due to the hunting regulations, we did not have access to the youngest age categories where the higher RVA circulation is expected. Similar to domestic pigs, age and gender were not significant factors for RVA prevalence.

Genetic diversity in domestic pigs was high in both genomic segments, with eight different G (G1-G6, G9 and G11) and seven different P genotypes (P[6]–P[8], P[11], P[13], P[23] and P[32]). In wild boars, the RVA strains were less genetically diverse with five detected G (G3, G5, G6, G9 and G11) and one detected P genotype (P[13]). The genotyping protocols were more challenging for wild boar samples, similar to what we previously reported in another wildlife species, i.e., red foxes [20]. The impact of the low RVA genomic concentrations (63.4% of RVA positive wild boars with Cq > 32), and the presence of diverse RVA strains influencing primer specificity, should not be excluded. Therefore, the underlying RVA genetic diversity might be even higher. On the other hand, the higher genetic diversity of RVA strains that was discovered in domestic pigs bred on large holdings, compared to the small backyard holdings, was predictable due to the more frequent animal movements in that type of holding.

The most dominant G genotypes found to be circulating in domestic pigs during the course of the study were G9, G5 and G3 and they account for 68.7% of all sequenced RVA strains. Considering the P genotypes, only two genotypes, more precisely P[13] and P[23], equaled 82.1% in total. All these genotypes are common in domestic pigs with varying spatio-temporal prevalence [3]. Previous reports indicate the substantial dominance of the G5P[7] genotype combination in domestic pigs worldwide [3]. Our results report a changing pattern, with G5P[13] and G9P[23] being the most abundant with 49.6% of all detected genotype combinations (Figure 2C). The combination G9P[23] was recently reported to be among the most frequent in Germany [46] and Spain [15]. Apart from the remarkable genetic diversity of each segment (VP7/VP4), we observed a striking number (*N* = 23) of different genotype combinations, higher than previously reported data for four countries combined (*N* = 21) [10], but lower than what was previously reported in Poland (*N* = 33) [47]. Moreover, a distinguished intragenotype diversity, i.e., the circulation of several lineages (genotypes G3-G5, G11, P[6] and P[13]) or the existence of possibly novel lineages (genotypes G2, G3 and P[32]), was further observed. The orientation of our study to locally bred domestic pigs, excluding the holdings with imported weanling and fattening pigs, might have contributed to these results. Nevertheless, the import of domestic pigs, which is common in Croatia, primarily from Western European countries, still has an important impact on animal health and the introduction of novel viral pathogens or certain genomic variants [48].

Among other less prevalent genotypes, the most interesting finding is the emergence of G1P[8] strains (*N* = 7), which are commonly observed in humans [49]. Both genotypes which were found in domestic pigs clustered within typical human lineages (Figure 3A and Figure 4A) and were detected during the last sampling season (2020/2021) in several holdings in three counties. This is not the first time such possible reverse zoonotic events, including G1 and/or P[8] strains, have been reported in domestic pigs [45,47,50,51,52]. Notably, our investigation of other samples collected on these holdings revealed that G1 and P[8] strains were involved in reassortment events with other circulating RVA strains of different genotypes. However, it seems that the VP7 is more readily involved in such reassortment events compared to the VP4, since G1 was found in combination with three additional different P genotypes (P[7], P[11] and P[23]) in a total of six reassortant strains, compared to only one G3P[8] reassortant strain. A possible reason lies in the VP7 segment having the lowest host-species barrier of all RVA segments, despite the proposition that VP4 may be the segment that reassorts more frequently [53]. Therefore, it is not surprising that in the present study, G1 strains were detected more often (*N* = 19) in domestic pigs compared to the P[8] strains (*N* = 8), indicating possible enhanced host adaptation. Most certainly, the host adaptation was further driven by reassortment with the already present genotypes. Such assumptions need to be further investigated on a larger sample set by implementing multiplex RT-PCR and whole genome sequencing. Sanger sequencing applied in the present study may underestimate the presence of certain genotypes in mixed infections. Therefore, the combination of different approaches may result in more certain conclusions.

The porcine-to-human spillover was often reported for the G4 and P[6] strains [40], which were not among the highly prevalent RVA genotypes within the present study. Nevertheless, their close phylogenetic relatedness with different referent porcine-like RVA strains in humans (Figure 3B and Figure 4A) speaks in favor of their zoonotic potential.

The interesting finding of the current study is the circulation of the P[32] genotype, which was first detected in Ireland [54] and Denmark [10] and further in the UK [50], Germany [46], Poland [47] and Switzerland [55]. Even though it was confirmed circulating in all three sampling seasons (Figure 2), it was restricted to only the two northernmost counties (Figure 1) and to only five samples, indicating its regional potential. These strains were rather distantly related to other available P[32] strains from the GenBank (<89% nucleotide sequence identity), which may indicate the circulation of this certain lineage for some extended period of time within the area. Continuous monitoring of RVA strains in domestic pigs, which is generally not present, would provide more information on the importance of such underrepresented genotypes.

Apart from possible interspecies transmission with humans (G1P[8]), we identified a small number of samples with G6 and P[11] genotypes, implying interspecies transmission events with bovines, which is also evident from their phylogenetic clustering (Figure 3B and Figure 4A). Both genotypes were detected as reassortants with typical porcine G3 and P[13] genotypes. Similar findings were reported elsewhere [3], confirming the existence of similar, but sporadic porcine–bovine reassortant strains.

Nevertheless, the most prominent number of interspecies transmission events in the current study was observed between domestic pigs and wild boars. More precisely, all genotypes detected in wild boars (G3, G5, G6, G9, G11 and P[13]) were detected in domestic pigs as well. The evidence that supports the statement that the natural transmission of RVAs between these two species actually occurs is the high sequence identities and phylogenetic relatedness depicted in Figure 3A,B and Figure 4B. Most of these RVA genotypes (G5, G9, G11 and P[13]), which we discovered circulating in the wild boar population, were already confirmed relevant for that species [16,17]. The exception was primarily the G3 genotype, which was detected for the first time in wild boars and it was in fact the most dominant genotype within the present study. Considering that the G3 genotype was the third most prevalent genotype in domestic pigs, it is not a surprise. Moreover, the RVA strains of the G3 genotype are believed to have the broadest host range [56]. The extent of inclusion of these two species in the natural transmission of certain RVA genotypes was observed by the previous detection of G4, P[6] and P[23] genotypes in wild boars in the Czech Republic and Japan [16,17] and, additionally, the G1 and the P[7] in the Czech Republic (unpublished genotypes available in the GenBank) (Figure 3A and Figure 4A).

An interesting observation on how porcine RVA strains impact the genetic heterogeneity of RVAs retrieved from another, yet distant, wildlife species was described in our previous investigation conducted on red foxes [20]. When fox RVA strains were compared to strains derived from the current study, it was evident they share porcine-related G9, G11, P[13] and P[23] genotypes with the prominent phylogenetic relatedness for the three most common genotypes (G9, P[13] and P[23]) (Figure 3B and Figure 4B). Since both species, wild boars and red foxes, tend to enter rural and urban areas, their contact with different animal species and their pathogens is expected. The contact of wildlife with domestic pigs and their manure is foreseen, especially in the matter of small backyard holdings, which usually have low biosecurity conditions.

In conclusion, our results contribute to the basic knowledge of RVA prevalence, genetic diversity and molecular epidemiology and to the extent of interspecies transmission events in domestic pigs and wild boars. Such baseline data may be considered important for the development and introduction of RVA vaccines in domestic pigs, an essential tool for pig health management. This is especially valuable for countries in the European Union where currently no authorized RVA vaccine for domestic pigs is commercially available. Lastly, the continuous monitoring of RVA in different species allows the prompt detection of new emerging variant strains that could become important for human health and the future effectiveness of vaccines currently in use.

## Figures and Tables

**Figure 1 viruses-14-02028-f001:**
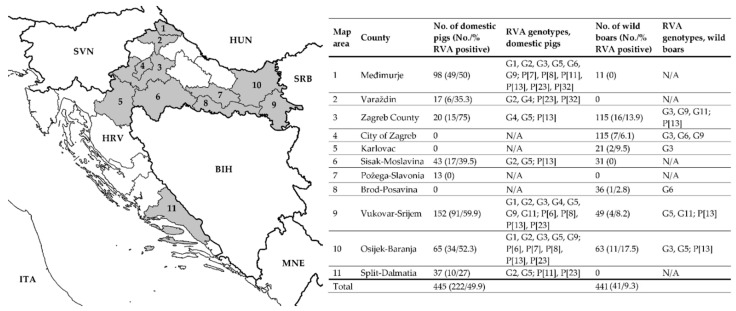
Geographical distribution of sampling sites and the results of RVA detection and genotyping (VP7 and VP4) in domestic pigs and wild boars. Counties included in the present study are marked in grey. N/A stands for Not Applicable in the case there were no samples taken or all collected samples were negative on RVA. The map source is available at: https://commons.wikimedia.org/wiki/File:Croatia_location_map.svg (NordNordWest; CC BY-SA 3.0; accessed on 11 January 2019).

**Figure 2 viruses-14-02028-f002:**
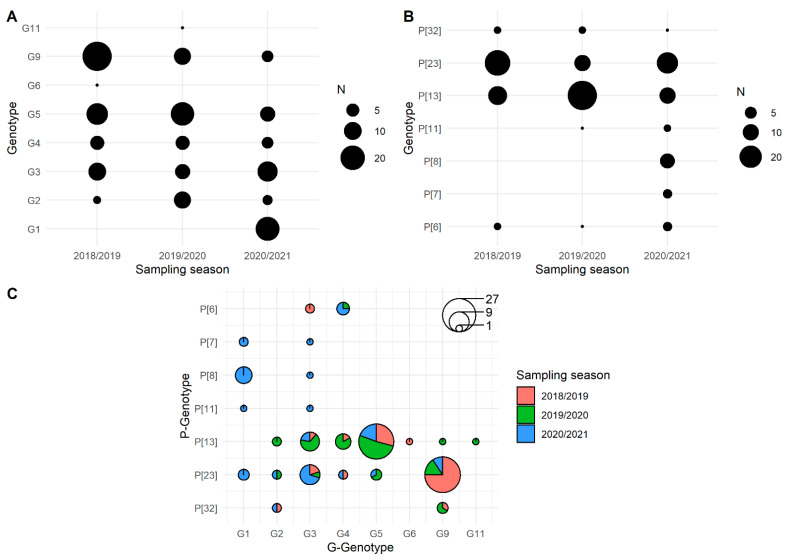
Temporal distribution of RVA’s G genotypes (**A**), P genotypes (**B**) and G/P genotype combinations (**C**) detected in domestic pigs. The circle sizes are proportional to the number of detected RVA strains.

**Figure 3 viruses-14-02028-f003:**
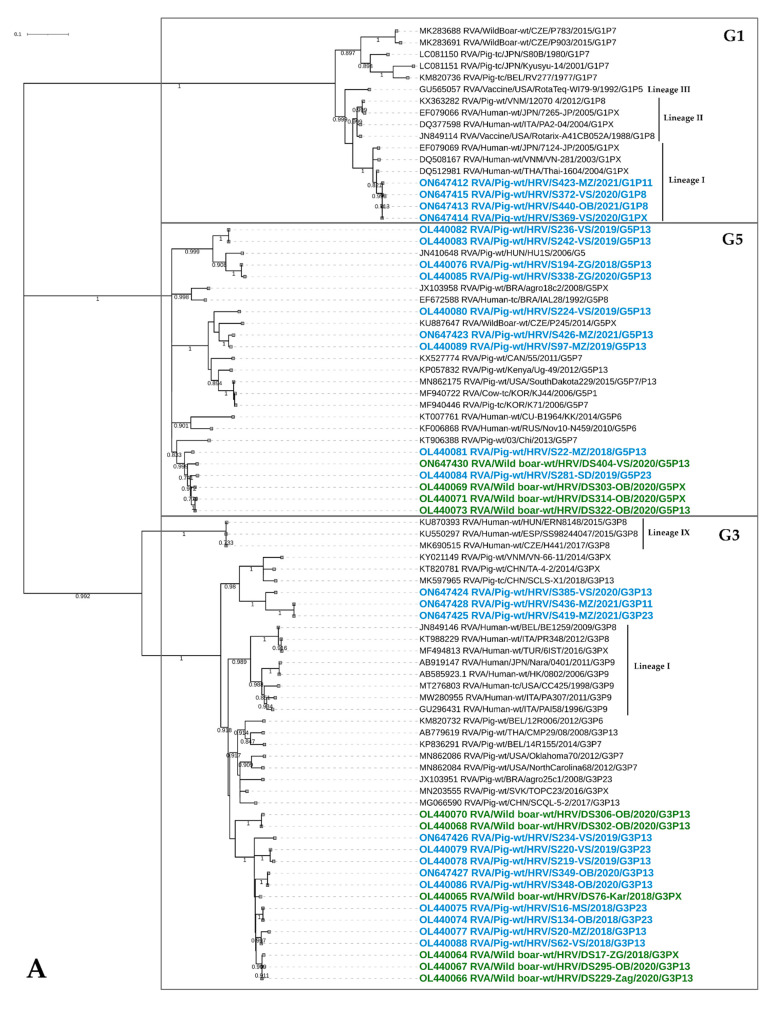
Phylogenetic relationship between RVA strains of G1, G3, G5 (**A**) and G2, G4, G6, G9, G11 (**B**) genotypes. The strains from the present study that were derived from domestic pigs and wild boars are marked in blue and green, respectively. The accession numbers of all strains, including referent strains from the GenBank, are designated within taxa. Based on the partial VP7 sequences (~800 nt), both trees were generated by the ML method and T92+G+I model in MEGA 11 software. The branching stability of each phylogenetic tree was assessed by 1000 bootstrap replicates (values indicated adjacent to the nodes if >0.7). The scale bar represents the number of substitutions per site. In displaying RVA strain nomenclature within taxa, the brackets for the P genotype were omitted for the sake of simplicity.

**Figure 4 viruses-14-02028-f004:**
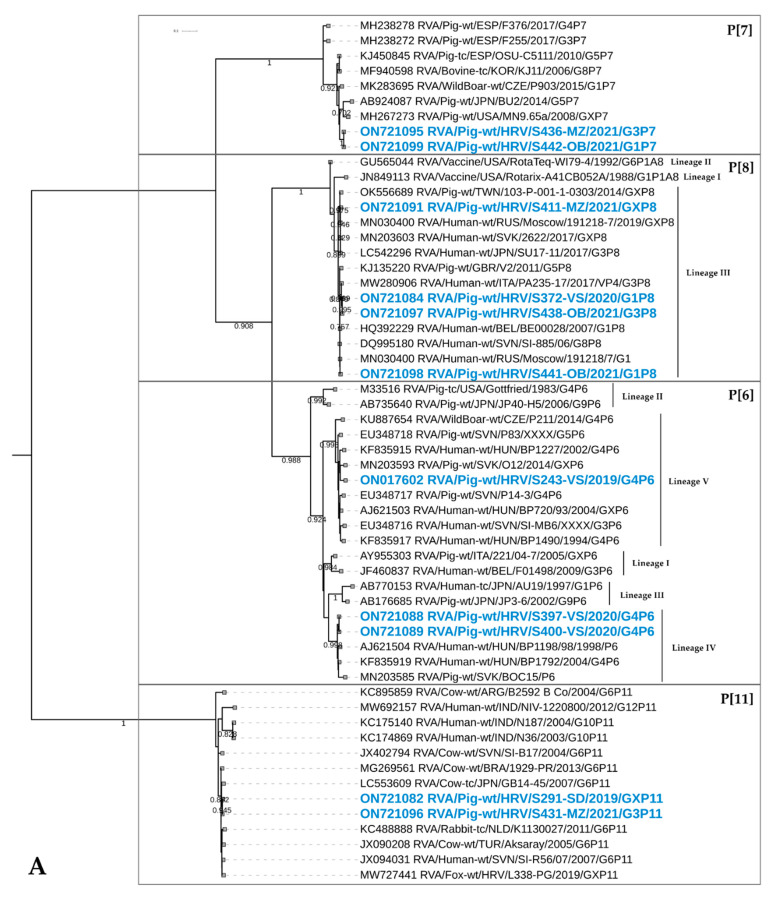
Phylogenetic relationship between RVA strains of P[6], P[7], P[8], P[11] (**A**) and P[13], P[23], P[32] (**B**) genotypes. The strains from the present study that were derived from domestic pigs and wild boars are marked in blue and green, respectively. The accession numbers of all strains, including referent strains from the GenBank, are designated within taxa. Based on the partial VP4 sequences (~650 nt), both trees were generated by the ML method and T92+G (**A**) or T92+G+I (**B**) model in MEGA 11 software. The branching stability of each phylogenetic tree was assessed by 1000 bootstrap replicates (values indicated adjacent to the nodes if >0.7). The scale bar represents the number of substitutions per site. In displaying RVA strain nomenclature within taxa, the brackets for the P genotype were omitted for the sake of simplicity.

## Data Availability

The datasets used and/or analyzed within the frame of the present study can be provided by the corresponding author upon a justified request.

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
