# Peer review of "Rotavirus A in Domestic Pigs and Wild Boars: High Genetic Diversity and Interspecies Transmission"

_viruses, 2022, doi:10.3390/v14092028_

Round 1
Reviewer 1 Report
Rotavirus surveillance among the domestic swine and wild boar populations is underexplored in comparison to this disease among humans. By carrying out geographical surveillance of rotavirus among domestic swine herd populations and wild boar populations, Brnić and team bring to light the importance of continued research in this area to prepare for the potential spillover of novel variants from wild boar into the domestic pig population. They also address the lack of a licensed rotavirus vaccine for use in domestic pigs in Europe and emphasize the importance of vaccination in highly prevalent regions.
Upon review of this article, I provide here my suggestions and my recommendation that will aid towards publication.
Minor Suggestions/Questions:
Introduction: Please write a few sentences regarding the current vaccination strategies/efforts towards the introduction of vaccination initiatives within the domestic swine population in Europe and why this initiative was not opted as compared to the US. This will provide the reader with a perspective on the current EU’s stance on vaccine development for domestic pig populations.
Please highlight the health-based outcomes of Rotavirus disease among domestic pigs. Rotavirus is relatively self-limiting among infected pigs and protection is conferred from sows to weanlings through maternal colostrum which is a cheaper method of maintaining herd health.
Line 39: “…especially among younger animals…” instead of “…especially for younger age categories…”
Please go through the article once more for general grammatical corrections.
Methods:
Lines 181-183: Please confirm the GenBank accession numbers as they are not accessible.
Reviewer 2 Report
This manuscript describes prevalence, genetic diversity and molecular epidemiology of RVAs from pigs and wild boars based on analysis of multiple samples collected in Croatia for three seasons. The study design, methodology and results in this study are unique, interesting and valuable on the research field for rotavirus. This study has high impact and scientific significance, and hence it is suitable for publication in this journal.
However, this manuscript has two minor points before publication.
Minor points:
Page 3, line 112-114
The authors have detected RVA from samples by real-time RT-PCR targeting VP2.
VP2 is classified into 24 genotypes.
Please show range (what kind of genotypes) of VP2 genotypes can be detected by this method.
Please discuss about limitation of this method.
Page 3, line128-134 and Line 135-140
The authors used three different methods to determine VP7 and VP4 genotypes, respectively.
Please discuss differences of detection rates and detection of specific genotypes by three different methods in Discussion section.
Reviewer 3 Report
Section 2.1: Please include a brief description mentioning that each pig was sampled once during each sampling season. I believe same pig was not sampled at multiple time points. This will make it easier for the readers to understand the study design.
Section 2.2, Line 115 – 123: I suggest to not include the PCR cycling details in the manuscript, if the method has been reported earlier. It is ok to include any changes or improvements made in the methods reported in the original publication. However, if no changes were made, then a reference to the original publication should suffice.
Section 3.1: Line 199-200: Please rewrite the sentence describing RVA prevalence. In its current form, the sentence reads more like a reference to a previous study than a result from the present study.
Section 3.1: Line 199-200: Please rewrite the sentence describing RVA prevalence. In its current form, the sentence reads more like a reference to a previous study than a result from the present study.
Section 3.1: Line 212: Please delete the sentence “which were mostly….previously noted”.
Section 3.1: Line 220-221: Please rewrite the sentence “The observed….were above 32”. Difficult to follow.
Section 3.2: Line 230: Replace “above” with “in materials and methods”.
Section 3.1: Line 250-253: Reference to sampling point per holding is not clear. Please see my comment on Section 2.1. I suggest to reword this sentence to bring more clarity.
Section 3.2, Line 172-175: If I understand correctly, samples from domestic pigs and wild boars were collected during three rotavirus seasons. I do not see any temporal distribution data for rotavirus A detected from wild boars. Although, only 5 G and 1P genotypes were detected from wild boars, authors can include the temporal distribution of RVA in wild boars in a bar chart or a table format. If the data is not available, then avoid mentioning RVA temporal distribution in wild boars anywhere in the manuscript (please see my comment on Section 4, Line 512-514)
Section 4, Line 497-498: I suggest to compare the RVA prevalence from this study to the studies available from other parts of the world. Authors have included a study from Spain but there are several studies available describing RVA in domestic pigs. For example, following studies could be included:
Marthaler D, Homwong N, Rossow K, Culhane M, Goyal S, Collins J, Matthijnssens J, Ciarlet M. Rapid detection and high occurrence of porcine rotavirus A, B, and C by RT-qPCR in diagnostic samples. J Virol Methods. 2014 Dec;209:30-4. doi: 10.1016/j.jviromet.2014.08.018.
Theuns S, Vyt P, Desmarets LMB, Roukaerts IDM, Heylen E, Zeller M, Matthijnssens J, Nauwynck HJ. Presence and characterization of pig group A and C rotaviruses in feces of Belgian diarrheic suckling piglets. Virus Res. 2016 Feb 2;213:172-183. doi: 10.1016/j.virusres.2015.12.004.
Pham HA, Carrique-Mas JJ, Nguyen VC, Ngo TH, Nguyet LA, Do TD, Vo BH, Phan VT, Rabaa MA, Farrar J, Baker S, Bryant JE. The prevalence and genetic diversity of group A rotaviruses on pig farms in the Mekong Delta region of Vietnam. Vet Microbiol. 2014 Jun 4;170(3-4):258-65. doi: 10.1016/j.vetmic.2014.02.030.
Section 4, Line 500: Please replace “method” with “method of detection”
Section 4, Line 502-503: The reference (Wu et al. 2022) mentioned in this line used a combination of end-point PCR and EIA for RVA detection mot just EIA. Please correct.
Section 4, Line 504-505: Please include a comparison of RVA prevalence in suckling and weaning piglets with other studies, if any. A quick search revealed a following recent publication describing RV prevalence in piglets of different ages.
Ferrari E, Salogni C, Martella V, Alborali GL, Scaburri A, Boniotti MB. Assessing the Epidemiology of Rotavirus A, B, C and H in Diarrheic Pigs of Different Ages in Northern Italy. Pathogens. 2022 Apr 14;11(4):467. doi: 10.3390/pathogens11040467.
Section 4, Line 512-514: I did not find temporal distribution data of RVA genotypes in wild boars in the manuscript. I suggest to either include the temporal data in the results or remove description of RVA temporal distribution in wild boars. More specifically, remove “which provides a ……..phylogenetic comparisons”.
Round 2
Reviewer 3 Report
Authors have done a good job in comprehensively revising the manuscript and responding to my queries. In my opinion the manuscript can be accepted in its current form.